# Advances in Silicon-Based Integrated Lidar

**DOI:** 10.3390/s23135920

**Published:** 2023-06-26

**Authors:** Mingxuan Hu, Yajun Pang, Long Gao

**Affiliations:** 1Center for Advanced Laser Technology, School of Electronic and Information Engineer, Hebei University of Technology, Tianjin 300401, China; hu010609@foxmail.com; 2Hebei Key Laboratory of Advanced Laser Technology and Equipment, School of Electronic and Information Engineer, Hebei University of Technology, Tianjin 300401, China; 3Beijing Institute of Space Mechanics & Electricity, Beijing 100094, China

**Keywords:** silicon-based Lidar, OPA Lidar, optical switch Lidar, FMCW, ToF

## Abstract

Silicon-based Lidar is an ideal way to reduce the volume of the Lidar and realize monolithic integration. It removes the moving parts in the conventional device and realizes solid-state beam steering. The advantages of low cost, small size, and high beam steering speed have attracted the attention of many researchers. In order to facilitate researchers to quickly understand the research progress and direction, this paper mainly describes the research progress of silicon-based integrated Lidar, including silicon-based optical phased array Lidar, silicon-based optical switch array Lidar, and continuous frequency-modulated wave Lidar. In addition, we also introduced the scanning modes and working principles of other kinds of Lidar, such as the Micro-Electro-Mechanical System, mechanical Lidar, etc., and analyzed the characteristics of the Lidars above. Finally, we summarized this paper and put forward the future expectations of silicon-based integrated Lidar.

## 1. Introduction

Laser has the characteristics of high brightness, monochrome, good directivity, and low coherence. Since 1960, when T.H. Miaman developed the first laser transmitter [1], laser has become increasingly common in daily life. Combining laser with traditional radar, researchers produced the laser radar (Lidar) technology. Compared with the former, Lidar presents better confidentiality and higher resolution. Usually, the angular resolution of Lidar is no less than 0.1 mrad, the range resolution can be up to 0.1 m, and the velocity resolution can be up to 10 m/s, which enables the Lidar to draw clear 3D point-cloud images by scanning.

At present, the technology of mechanical Lidar is mature. It is composed of laser emitters and rotating parts. This kind of Lidar arranges the multi-line laser emitter in the vertical direction, emits the laser at a specific frequency, and realizes dynamic scanning combined with the uniform rotation of the rotating parts. Figure 1 shows mechanical Lidar from Velodyne with 64 lines (a) and 128 lines (b). The HDL-64E is 13.2 kg in weight which limits its application. Mechanical Lidar dominated the market for a long time due to its 360° horizontal scanning angle and simple way of working. However, this type of Lidar has some limitations. The necessary rotating parts lead to a large size, reliability issues, and high production cost [2].

Micro-Electro-Mechanical System (MEMS) Lidar uses the laser source to emit the laser beam and the tiny mirror inside the device vibrates at a specific frequency to reflect the laser beam to realize dynamic scanning [3]. Compared with mechanical Lidar, MEMS does away with bulky motors, significantly reducing the size of the mechanical components, reducing the weight of the device and decreasing production costs [4]. Figure 1c shows the product of Velodyne, which is based on MEMS with 120° × 16° FoV. MEMS Lidar is used in UAV platforms [5] and small robots [6]. In addition, The Scala GEN.1 Lidar from Valeo is the first mass-produced laser scanner for use in automobiles [7], as shown in Figure 1d. Due to the limited optical aperture, MEMS Lidar has a small field of view. However, this kind of device still has fragile mechanical parts and the issue of vulnerability to mechanical shocks is still there [8].

Solid-state Lidar that provides scalability, reliability, and embeddedness without mechanical parts has caused great interest [9]. Solid-state Lidar includes Flash Lidar, optical phased array, and optical switch array Lidar. Flash Lidar works like a digital camera, shining a large laser beam across a scanning area and capturing the backscattered light with a detector to work out the shape and position of the target. This method has a high scanning rate but a short detection range. Because the power to illuminate the whole FoV always is limited by eye safety issues [10], Flash Lidar is currently used in UAV platforms and image multiphase segmentation technology [11,12].

Silicon-based optical phased array Lidar integrates a laser emitter, beam splitter, phase modulator, and other electronic devices on a tiny platform of several square millimeters, significantly reducing the device’s size. In addition, since the photoelectric integrated circuit is fully compatible with the existing and mature CMOS process [13], the production cost of silicon-based Lidar is further reduced. Compared with mechanical and MEMS Lidar, optical phased array Lidar uses a phase modulator to separately adjust each laser beam. The laser beam passing through the phase modulator will have constructive interference in the specified direction and destructive interference in the other directions, resulting in a high-intensity beam in the direction of phase interference. The intensity of the radiation in the other directions is close to zero, which controls the direction of the laser scans. In this way, longer and higher-resolution ranging can be realized by increasing aperture size and array size or optimizing the arrangement of elements on the chip [14,15].

In addition to the optical phased array, there is also a silicon-based optical switch array Lidar technology is gradually being paid attention to by researchers. The optical switch is a vital device in optical integrated circuits or optical networks. It can be applied in optical communications and optical sensors such as Lidar. The optical switch Lidar often adopts the Machzender interferometer (MZI) structure. The MZI was proposed by Zehnder and Mach in the 1890s. The structure splits a beam of light into two beams that pass through their own modulation channels and then merges them into a single beam. The greatest advantage of the MZI is that it is not sensitive to temperature [16]. Therefore, it can be well adapted to the thermal phase shift working mode with low loss [17,18]. The optical switch Lidar with the MZI structure can change the phase shift type according to the requirement. For example, the thermo-optic effect has a small insertion loss but extra power consumption; and the electro-optic effect has a fast switching speed [19].

In summary, various types of Lidar have different advantages and disadvantages. Table 1 summarizes the differences between the above types of Lidar. We can choose the type of Lidar according to the work requirements and estimated cost. Next, the research progress of silicon-based integrated Lidar is introduced in detail.

## 2. LIDAR Ranging Methods

Lidar mainly uses the three methods of triangle ranging, time of flight (ToF), and frequency-modulated continuous wave (FMCW) to scan the ranging of the surrounding objects.

### 2.1. Triangular Ranging

The specific principle of trigonometry [20,21] is shown in Figure 2.

After the laser is emitted by the transmitter in Figure 2, it is reflected or scattered by objects at different positions. Part of the reflected light is imaged on the detector, such as a charge-coupled device (CCD), after passing through the receiving lens. Objects at different positions are imaged at different positions on the CCD. Combined with the geometry knowledge of similar triangles, the distance between the actual object and the laser emitter can be calculated by the spot distance on the CCD.

The Lidar technology of triangulation ranging is relatively mature, which has a significant advantage in production cost. Moreover, thanks to the parallel axis optical path arranged by triangulation ranging, the overall appearance of the Lidar is relatively low, which is suitable for the cruise assistance of a sweeping robot at present. However, the triangular ranging method is unsuitable for far-distance ranging because the farther the distance, the closer the spot spacing, and the more difficult it is to distinguish. Therefore, Lidar based on this method is mainly used in consumer products and application in the industrial field is limited.

### 2.2. Time of Flight

The Time of Flight (ToF) ranging principle is shown in Figure 3.

The light source emits a separate high-power pulse and the time measurement circuit is triggered instantly [22]. When the pulse is reflected off the target and detected by the photodiode, the timer circuit measures the time it takes the pulse to travel from the transmitter to the target and finally to the photodiode. We can calculate the distance from the target to the Lidar based on the measured time [23]:(1)$$R=\frac{c\cdot ∆t}{2},$$
where c is speed of light in air. Alternatively, the light source can emit continuous waves [24,25] and we can calculate the time indirectly by measuring the phase shift of the received signal. Time of flight is applied in the deep camera to obtain more space information from the photos [26].

**Figure 3 sensors-23-05920-f003:**
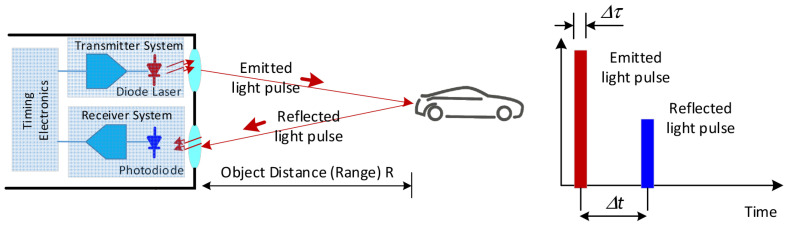
The principle of Time of Flight [23].

The principle of ToF is straightforward, but engineering implementation has many difficulties. For example, it needs to emit a very high-power optical pulse signal and the receiver needs to be very sensitive to receive weak reflected signals. Because the speed of light is breakneck, the circuit delay must be strictly controlled to measure the precise time value of the optical pulse sent to the reception in order to calculate the accurate distance.

### 2.3. Frequency-Modulated Continuous Wave

The major advantage of FMCW Lidar lies in its ability to measure the velocity information [27]. The Lidar working in FMCW mode transmits frequency-modulated continuous lasers, and the laser is split into two beams. One beam is emitted to the target while another is fed directly onto the photodiode PD as the reference signal (or local oscillator) [28,29], which compares the sample signals with the reference signals in the time domain and frequency domain, then calculates the distance and velocity of the measured object according to the Doppler effect.

The sample signal and the reference signal have a beat frequency generated by the temporal delay between the two signals. For the static target, the beat frequency can be expressed as [30]:(2)$$f_{reference}-f_{sample}=f_{b}=\frac{2B}{Tc}R,$$
where $f_{reference}$ and $f_{sample}$ represent the frequency of reference and sample light, $f_{b}$ represents the beat frequency, B is the optical frequency sweep range of the laser, T is the sweep period, c is the speed of light in the air, and R is distance between the target and the FMCW Lidar. However, for a moving target, the beat frequency is not constant because of the Doppler effect. There will be a Doppler frequency generated by this effect. We can figure out the velocity by the formula:(3)$$v=\frac{f_{shift}\cdot \lambda }{2cos\theta }=\frac{f_{shift}\cdot c}{2cos\theta \cdot f_{0}},$$
where v represents the instantaneous velocity of the moving target, f_shift_ is the Doppler frequency, λ is the wavelength of light, θ is the angle between the target velocity vector and the beam path, and f_0_ is the center optical frequency of the laser. The beat frequency is different due to the Doppler frequency. As shown in Figure 4b, during the up-chirp, the actual frequency shift can be expressed as:(4)$$f_{b,uc}=f_{b}-f_{shift},$$
while in the down-chirp, it is expressed as:(5)$$f_{b,dc}=f_{b}+f_{shift},$$
and we can figure out the Doppler frequency and beat frequency by the formula:(6)$$f_{b}=\frac{f_{b,uc}+f_{b,dc}}{2},$$
(7)$$f_{shift}=\left|\frac{f_{b,uc}-f_{b,dc}}{2}\right|.$$

Generally, the beat frequency of the target approaching or receding the Lidar will decrease or increase linearly over time. However, for the vibrating object, the beat frequency will change in a sine curve and the phase difference between the f_b,uc_ and f_b,dc_ as shown in Figure 4c. According to this, we can figure out the vibration frequency using FMCW Lidar [30].

## 3. Silicon-Based Optical Phased Array Lidar

Mechanical Lidar is often limited in performance and cost due to mechanical beam orientation and stabilization mechanisms. In order to achieve random pointing and stable scanning results, the Lidar needs to rotate precisely and quickly, leading to high power consumption. To solve this problem, researchers proposed a Silicon-based optical phased array Lidar with low power consumption and a stable working state. The principle of the optical phased array is similar to that of the microwave phased array. The direction of the main beam can be changed by controlling the phase difference between the beams. The Lidar formed by an optical phased array completely gets rid of mechanical components.

An optical phased array is usually composed of a coherent light source, beam splitter, phase shifters, and optical antennas [31]. The light source is divided into multiple waveguides by the splitter, then modulated by the phase shifters and transmitted to the free space by the grating couples. The time taken for the light to reach the equiphase surface is the same when the phase of each ray is the same. Suppose there is a uniform phase difference between two adjacent paths. In that case, the equiphase surface is no longer perpendicular to the direction of the original path but produces a beam deflection angle. According to the knowledge of optical interference, the optical path of the equal phase will be longer than that of the coherent phase, and the optical path of the unequal phase will cancel each other. Therefore, the final synthesized wavefront forms a certain angle with the propagation direction of the original optical path [32].

Usually, 2D beam steering can be achieved in two ways as shown in Figure 5: (a) in a one-dimensional optical phased array, the beam steering is accomplished by wavelength tuning in the longitudinal direction, θ, and phase shifting in the lateral, φ. The phase difference between adjacent grating couplers is ϕ; (b) in a two-dimensional optical phased array, the deflection of the beam in both directions is realized by phase delay. Some parameters of the OPA Lidar have been listed in Table 2.

### 3.1. OPA Lidar Based on Silicon on Insulator

As early as 1997, P. Trinh et al., University of California, produced the first four-channel wavelength multiplexer fabricated in the silicon-on-insulator [51]. The channel interval of the wavelength multiplexer was 1.9 nm, the crosstalk of the adjacent channel was less than −22 dB, the on-chip insertion loss of the channel was less than 6 dB, and the TE-TM shift was less than 0.04 nm, which reached the minimum polarization displacement in the waveguide circuit without compensation technology at that time. However, the scanning angle is limited to the chip plane.

In 2005, F. Xiao et al. theoretically proposed a non-uniform phased array [52] to control the beam by changing the wavelength. This method can effectively suppress the side lobe and reduce the difficulty of producing and controlling. After two years, F. Xiao et al. presented the theory of a cascaded irregular phased array [53]. They calculated the relationship between the device size and spacing in each array stage and realized the beam’s phase matching at the two-stage cascade. Based on the above simulation and theory, this research group produced 16- and 32-element irregular phased arrays on silicon-on-insulator materials in 2008 [33]. The device consists of irregular elements that allow a continuous sweep range of 0° to 3° when the wavelength varies from 1550.7 nm to 1551.9 nm.

In 2009, K. Van Acoleyen et al., from the University of Ghent reported a one-dimensional optical phased array composed of 16 grating couplers spaced 2 μm apart [34], as shown in Figure 6. At a wavelength of 1550 nm, it can achieve continuous thermo-optical steering of 2.3° and wavelength steering of 14.1°. However, considering that excessive current will destroy electronic devices in actual use, the current is limited to 3.95 mA, so the transverse scanning range is also limited.

K. Van Acoleyen et al. reported a two-dimensional optical phased array fabricated on SOI in 2010 [35]. The input waveguide is connected to a grating coupler which couples light into the system. This grating coupler uses the diffraction principle to couple the light from a single-mode fiber into the TE-like mode of a 10 μm waveguide, and then the waveguide gradually shrinks to a 450-nm-wide photonic wire as the input waveguide. Finally, it is split into N × N output waveguides by the MMI tree. This method achieves 0.24°/nm wavelength steering, eliminates the need for an active phase modulator, and reduces the antenna loss to less than 3 dB.

In 2011, J.K. Dylend et al., from the University of California, demonstrated a 16-channel, independently tuned waveguide surface grating optical phased array [36], which can realize a two-dimensional scanning field of view of 20° × 14°. In addition, the scheme uses infrared cameras to record real-time far-field images for feedback, which minimizes phase error and background peak noise. The far-field beam width is 0.6° × 1.6°, the far-field angular resolution is less than 1°, and background peak suppression is 10 dB.

In the same year, D. Kwong of the University of Texas proposed to make a 12-channel optical waveguide phased array by using a silicon nanomembrane with an off-chip laser as the light source [37]. Individually controllable micro-heaters modulate the optical phase and achieve a free space scanning angle of 31.9° at a wavelength of 1550 nm. The phased array uses a non-uniform waveguides array, which not only restrains the side lobe but also effectively prevents the coupling effect between adjacent waveguides and reduces the strict requirement of waveguide spacing when the beam deflects at a large angle.

In 2011, K. Van Acoleyen et al. proposed a one-dimensional beam scanning device based on the silicon-on-insulator [38]. The schematic diagram of the device is shown in Figure 7. The beam deflection is realized by the one-dimensional optical phased array composed of 16 waveguides. The waveguide width at the S-shape is 800 nm. A wider waveguide width can allow for fabrication deviation and reduces the corresponding phase error in the waveguide. At the bend, the waveguide becomes a single mode line with a width of 450 nm, which allows sharp turns of less than 3 μm without significant damage. This S-shaped waveguide structure keeps all channels in phase and allows for large waveguide spacing during fabrication. The device can achieve a 23° field of view angle and, combined with the star coupler, reduces the side lobe level by 13 dB.

Subsequently, in the same year, this research group produced an integrated two-dimensional beam-scanning device using a star coupler on an insulating silicon wafer [39]. The structure of the device is shown in Figure 8. In the ϕ direction, a high-order grating composed of Arrayed Waveguide Grating (AWG) enables the beam to sweep rapidly in a specific range. In the θ direction, a low-order grating is used to scan the beam slowly in a specific range and couple the light off the chip. In this way, the device can achieve a two-dimensional scanning field of view of 15° × 50° within the range of wavelength change in 100 nm. The beam width is around 4°. Furthermore, the beam width and scanning angle can be further optimized by changing the grating structure and component size. In addition, the coupling efficiency could be improved by adding another layer of polysilicon to the waveguide layer structure during fabrication [54].

In 2013, J. Sun et al., from MIT reported a two-dimensional large-scale nano-antenna phased array fabricated on a silicon wafer [55]. The research group integrated 64 × 64 (4096) optical nano-antennas densely on a 576 μm × 576 μm silicon wafer, which was the densest silicon-based optical phased array at that time. The system’s superior robustness and compatibility with CMOS technology enable all nano-antennas to be integrated on such a chip size. The schematic diagram of the system is shown in Figure 9. The laser can be input from the fiber and transmitted evenly to each nano-antenna by the waveguide. The coupling efficiency varies with the length L_c_ of the directional coupler. The two segments of the optical delay line can achieve a desired phase delay φ_mn_ to control the emit phase. The results show that, theoretically, any far-field radiation pattern can be realized by controlling the emission phase of all pixels. In practice, however, providing an accurate emission phase for every pixel is almost impossible because minor manufacturing errors can cause serious phase errors.

In 2014, A. Yaacobi et al., from MIT reported a high-speed, low-power, and wide-scan-angle optical phased array [40]. This phased array is based on a novel phase-shifting architecture that utilizes 32-μm-long grating-based antennas, fed through evanescent field waveguide couplers from a bus waveguide in directly integrated thermo-optic phase shifters. The antenna gratings were designed using a shallow etch on the sides of the waveguide to ensure uniform emission from the decaying field. A laser with an eye-safe wavelength of 1550 nm was used for demonstration. Under the conditions of 10.6 V voltage and 18 mW average power, a continuous steering range of 51° was achieved. In addition, the device can also adapt to the wavelength range of 1.2 to 3.5 μm.

In 2016, C.V. Poulton et al., from MIT produced a phased array composed of 50 grating-based antennas [41]. The device achieves a 2D beam steering range of 46° × 36° by thermal and wavelength tuning, and the minimum beam spot is 0.85° × 0.18°. A schematic diagram of the device is shown in Figure 10. The phase shifters are cascaded into three groups. This design allows phase fine-tuning to compensate for noise caused by manufacturing errors. In addition, the cascade structure also allows three sets of phase shifters in the device to operate under different voltage conditions to generate three beams with different directions. In this research group, the antenna was etched on the edge of the waveguide using full etching, as shown in Figure 10b. This method can reduce the negative impact on the transmitting antenna caused by transition etching or insufficient etching of the waveguide.

In 2016, D.N. Hutchison et al. reported a large non-uniform emitter spacing optical phased array, fabricated in a 300 mm CMOS facility [15]. Due to the non-uniform arrangement of emitters, this array can suppress the side-lobe effect and improve the field of view. The device can achieve a beam steering Angle of 80° by thermo-optical phase modulation and a beam steering angle of 17° by wavelength tuning. This is equivalent to over 60,000 resolvable points in two-dimensional beam steering. In addition, a large number of phase-controlled emitters were used to control the average beam divergence at 0.14° (the minimum value was 0.11°).

In 2019, C.V. Poulton et al. reported a 512-path optical phased array [42]. They set up phase shifters for each path that can be controlled independently, which makes the transmitted wave front more flexible. The power consumption of the device is reduced to 1 mW. The longer antenna can increase the directivity of the emitter. A large scanning range of 56° × 15° can be achieved. The device uses a wideband photoelectric phase shifter for phase control. Compared with the traditional thermal optical phase shifter, this method not only reduces power consumption but also limits the crosstalk between adjacent elements. In addition, the OPA chip developed by this research group was applied to the Lidar for the first time, and the scanning experiment of 25 m was demonstrated. The real-time scanning image as shown in Figure 11 presents clear character contour features at 6 m.

In 2019, P. Bhargava et al. presented the first integrated coherent Lidar system with experimental ranging demonstrations operating within the eye-safe 1550 nm band [43]. Due to the tight pitch (2 μm) between antennas, the antennas can achieve high gain and strong rejection of interfering beams. Germanium is also used to construct a photoelectric detector with high response speed that can work within the safe range of human eyes. This device proves that high-performance solid-state Lidar can be realized at a low cost.

In 2020, S.A Miller, from Columbia University, reported a multi-pass 512-channel optical phased array [44], as shown in Figure 12a. The device adopts a multi-pass platform composed of a multi-mode waveguide embedded with a phase shifter, as shown in Figure 12b. This platform can cycle the light several times in the channel. During the cycle process, the beam will have orthogonal space conversion of different modes, so as to eliminate unnecessary interference and maintain broadband operation. The beam circulates in the channel and accumulates phase shifting, thus realizing the beam steering. In this way, the power consumption of large-scale optical phased arrays is reduced by nearly nine times while maintaining low loss across at least 100 nm of continuous optical bandwidth. It is demonstrated that the optical phased array can achieve 70° × 6° beam scanning under the condition of 1.9 W power consumption. The overall package is shown in Figure 12c. The size of the phased array chip is 8 mm × 15 mm.

In 2022, S. Zhao et al. proposed a polarization multiplexing OPA with a bi-directional shared grating emitter array [45]. The system is composed of two phased arrays, a group of transmitting antenna arrays, multiple Mach-Zehnder interferometers (MZIs), and polarization factors (PSR). As shown in Figure 13, two identical OPAs are placed on either side of the transmitting antenna. MZIs and PSR are used to select the polarization state and propagation direction of the beam from the input to the transmitting antenna. The structure of the device improves the deflection range of the beam in the longitudinal direction. Through the test, this OPA can achieve longitudinal deflection of 54.5° by changing the wavelength from 1500 nm to 1600 nm when wavelength modulation efficiency is 0.545°/nm. Combined with phase modulation, the device can achieve a field of view angle of 54.5° × 77.8°. By increasing the number of antennas, the wavelength tuning range could be further improved. In addition, this OPA chip’s fabrication tolerance is high, reducing the fabrication difficulty.

### 3.2. OPA Lidar Based on Hybrid Material

In 2018, M. Zadka et al., from Columbia University, showed a hybrid silicon/silicon-nitride grating platform that overcomes the traditional trade-off between beam divergence and field of view [46]. Compared with Si-SiO_2_ waveguides, this platform is not sensitive to variations. Moreover, an 8 nm Al_2_O_3_ atomic layer is placed between the SiN and Si layers to serve as an etch-stop layer, thick enough to stop etching and thin enough to not disturb the entire device. The research group extended the length of a single antenna to 1 mm, increased the number of effective gratings, and made the beam divergence angle as low as 0.089°.

W. Xie, from the University of California, Santa Barbara, reported a high-density OPA system with heterogeneous phase shifters in 2019 [47]. The III-V PN diode is integrated into the silicon waveguides, and the diodes provide phase tuning for the optical mode in the hybrid III-V/Si waveguides. Combined with a star coupler and small spacing optical antenna, the phased array minimized the beam width at 0.02°, wavelength modulation efficiency reached 0.138°/nm, and a field of view of 51° × 28° was achieved. In addition, the device’s phase-shifter power consumption is extremely low, with static power consumption of less than 3 nW, wide optical bandwidth of more than 200 nm, and high operating speeds of more than 1 GHz. The chip composed of an optical phased array and a transceiver can further reduce the cost of Lidar and promote the development process of fully integrated Lidar.

In 2020, P. Wang et al., from the Institute of Semiconductors of the Chinese Academy of Sciences, proposed a SiN-Si dual-layer OPA chip [48]. The SiN layer is located above the SOI substrate with a spacing of 150 nm. The silicon devices and the SiN devices are located on two layers and do not interfere with each other, as shown in Figure 14. The input coupler and cascade beam splitter at the front end of OPA is a SiN device. The phase modulator and optical antenna at the back end are silicon devices. The SiN-Si chip can reduce the optical loss caused by the nonlinear effect when the silicon-based OPA works with high input power. At the same time, the front-end SiN device can handle very large optical power and is suitable for long-range detection.

In 2021, C.S. Im et al. proposed an optical phased array with silicon nitride mixed polymer integration [49]. Unlike the traditional silicon nitride phased array, this polymer has a high thermal and optical effect. The device integrates the polymer phase modulator with the silicon nitride power distributor and antenna to solve the problem of low phased array modulation efficiency of silicon nitride materials. In addition, a steering angle magnifying lens is used to improve the modulation efficiency of the antenna. The test results show that the modulation power of each channel of the optical phased array is P_π_ = 2.5 mW, and the field of view angle of 12° × 30° can be achieved. Finally, the research group combined this OPA to further produce the 3D Lidar based on ToF, and realize the scene scanning within 10 m. As shown in Figure 15, it is evidently found that a person at a distance of 8 m is distinguished from a wall at a distance of 10 m.

In the same year, L. Zhang et al. reported a multi-layer two-dimensional long-distance scanning optical phased array combining SiN and Si [50]. This device consists of a SiN base coupler, a SiN base beam splitter, and a Si base phase shifter. The beam is first coupled into the 200-um adiabatic waveguide by the mode-spot converter, then it is divided into 64 channels by the 6-stage Y beam splitter. Finally, the beam is transferred into the Si waveguide through the conical coupler. The device combines the advantages of the high-power processing capacity of SiN waveguides and the high thermal and optical modulation efficiency of the Si waveguide to show excellent performance. This OPA can realize the deflection angle of 96° × 14.4°. In addition, the OPA chip is used to successfully achieve long-range detection with up to 20 m distance in the ToF system and another test with a range of up to 109 m in the FMCW system.

From the above research progress of OPA Lidar, it can be found that researchers in various countries are striving to study the direction of miniaturization, low power consumption, and large field of view of Lidar. Despite the rapid development of OPA Lidar, some problems still need to be solved.

The problem of the side lobe is unavoidable for the on-chip Lidar made by the Fraunhofer diffraction effect, that is when the spacing of each incident waveguide is the same and larger than half wavelength, the side lobe that affects the scanning quality will appear. There are many methods to suppress the side lobe, among which the use of a non-equidistant waveguide phased array is relatively common. In 2016, Komljenovc et al. studied the non-equidistant waveguide phased array and proposed that the tiling method could realize the large-scale phased array with side-lobe suppression. In this paper, some researchers have also designed side lobe suppression schemes, such as Xiao, using non-uniformly distributed optical waveguide devices to suppress side lobe generation, and Hutchison, using a phased array composed of a star coupler to effectively suppress the side lobe when the distance between waveguides is greater than half wavelength.

The scanning angle is still limited. Among all the OPA Lidar mentioned above, the maximum scanning field of view that can be achieved is 77.4° × 28.2°. Achieving a larger scanning field of view means increasing the difficulty of the production process and increasing the device cost. The simplest way to increase the scanning angle is to use a lens, which prevents the overall device from miniaturizing and reduces its stability. In addition, improving the efficiency of thermo-optical modulation can also achieve a larger scanning angle. For example, the double spiral structure waveguide proposed by Adam Densmore above reduces the volume and improves the efficiency of thermo-optical modulation.

## 4. Silicon-Based Optical Switch Array Lidar

An optical switch array can realize the conversion of light beams from any input port to any output port. It is the most basic and core device in optical switching networks. It saves the process of optical-electrical and electrical-optical switching, so that the working mode is simpler and the corresponding switching speed is faster. Compared with an optical phased array, an optical switch array has the characteristics of small size, low power consumption, and high stability. In addition, the optical switch array is compatible with CMOS technology, so it is easy to make high-integration optical devices. Compared with silicon-based phased array Lidar, this kind of Lidar is more straightforward and more cost-effective. Table 3 describes some parameters of the optical switch array Lidar in this section.

### 4.1. Low-Consumption Optical Switch Array Lidar

In 2009, A. Densmore et al. designed an optical switch array device that can reduce thermal and optical power consumption by taking advantage of the small bending radius realized by silicon photonic waveguides [56]. Compared with the traditional waveguide arrangement, the team constructed the waveguide into a double spiral structure, allowing the device to extend the waveguide length to achieve a larger range of phase migration without increasing the volume of the heating parts. This method not only reduces the power consumption of the thermo-optical switch but also reduces the volume of the whole device. In addition, the device does not need to minimize the width of the heater to achieve lower power consumption, increasing the manufacturing tolerance of the heater. The test results show that a low switching power of 6.5 mW was obtained for a spiral-path Mach–Zehnder interferometer device with a 10–90% rise time of 14 µs.

In 2018, C. Chaintoutis et al., from the University of Athens, proposed to realize two-dimensional steering of light beams by means of wavelength tuning and piezoelectric transducers [57]. In order to minimize the divergence of the pixel-transmitted beam, the research group added a lens system on this basis, that is, a cylindrical lens was added in front of the chip to ensure beam collimation for more than 20 m, as shown in Figure 16. The beam steering in the horizontal direction is accomplished by the phase shifter relying on piezoelectric transducers (PZT) with low power consumption (μW) and high response time (ns). It is proven that the system can meet the requirements of low power consumption and low delay.

In 2019, C. Li et al., from Shanghai Jiaotong University, proposed a 4 × 4 antenna array based on an optical switch array [58]. They placed the lens on the top of the chip to focus the parallel light emitted from different emitters on the other side of the lens, that is, to realize the beam steering in discrete positions by letting the light be emitted by different emitters. The group demonstrated with 1550 nm light, and the results showed that the system can achieve two-dimensional steering and 19 dB background suppression. It is worth noting that this device has O (logN) power consumption for N emitters. It is a good way to decrease the consumption for large-scale emitter arrays.

In 2021, C. Rogers et al. demonstrated a large-scale coherent detector array with 512 pixels [59], as shown in Figure 17. The transmitter is composed of 1 × 16 thermal-optical switching trees. A heterodyne receiver with 32 × 16 pixels is used to detect and compare the received scattered light with the local oscillator light produced by the 1 × 8 switch array. The receivers are paired with the beam steering to illuminate the entire scene in small patches, thus eliminating the trade-off between the field of view and range. To obtain more pixels, they integrated a highly multiplexed array of electronic readout architecture into the receiver array, minimizing external electrical connections while maintaining signal integrity. Due to the coherent receiver array, the device can operate in FWCM mode, which has the advantages of strong anti-interference capability, velocity measurement, high precision, and suitability for optical integration, compared to ToF. The test results show that the measuring accuracy of the device can reach 3 mm under the laser power consumption of 4 mW.

### 4.2. Wide-FoV Optical Switch Array Lidar

In 2018, J.J. López et al. achieved a two-dimensional beam steering angle of 38.8° × 12° by using lens and wavelength tuning on the SiN platform [60]. The researcher placed the lens in front of the optical antenna as shown in Figure 18, and the optical switch array could change the angle of the light beam entering the lens to realize one-dimensional steering. The turning of the beam in the other direction is accomplished by wavelength tuning. Compared with the traditional phased array method, only a subset of the switches are used simultaneously, such that this scheme can save a lot of power when a large number of far-field points are needed. In addition, this approach reduces the complexity of the control architectures of both the transmitter and receiver, while also increasing their stability to temperature and environmental variations.

In 2020, H. Ito et al., from Yokohama University in Japan adopted a special prism lens that can suppress the dependence of collimation conditions on the angle of incident, and proposed a special shallow etched diffraction grating that can reduce emission loss, internal reflection loss, and collimation loss [61]. By combining the above two methods with a left-right symmetrical 1 × 16 optical switching network, they achieved 40° × 4.4° two-dimensional scanning, a beam divergence angle of 0.15°, and a number of effective spots diverging to 4256. Compared with OPA chips of the same performance, the device can be realized with lower power consumption and a simpler working mode.

In 2022, X. Zhang demonstrated a 16,384-pixel FMCW imaging Lidar with a monolithically integrated 128 × 128-element silicon photonic MEMS focal plane switch array (FPSA) [62]. Combined with a 5-mm-focal-length compound lens, this device can direct the light beam randomly to 16,384 distinct directions in a range of 70° × 70° with a 0.05 divergence angle and a microsecond switching time. As shown in Figure 19, each optical antenna is connected to a row waveguide by means of a MEMS optical switch (column-selection switch) that is connected to one of the input waveguides using a MEMS optical switch (row-selection switch). This way, the total number of control signals is reduced from N^2^ to 2N in an N × N FPSA. The laser can be routed to the corresponding antenna by turning on a row- and column-selection switch. Then, the emitted light is converted to a collimated beam by the lens.

The research of optical switch array Lidar started late and so far there is no production of optical switch array-based Lidar. The paper describes only some small-scale optical switching networks with limited FoV. Because the scale of an optical switch network directly determines the beam deflection angle realized by the chip, to reduce the optical loss caused by large-scale optical switch arrays, most optical switch arrays are combined with lenses to achieve a larger field angle. It can be seen that the optical switch array technology needs further research to meet users’ needs.

## 5. Integrated FMCW Lidar

At present, the mean ranging method of Lidar is ToF ranging. With the development of Lidar integration technology, it is likely that the products will eventually be commercialized and mass-produced. LuminWave believes that Lidar will be mass-produced at a low cost in the near future. It will be as common as cameras are today. It will be used in autonomous driving, smart city building, robotics, drones, and more, providing real 3D information directly as a core sensor for sensing the environment.

However, the popularity of Lidar will also cause problems. At present, most Lidar use the ToF method for ranging, which still has some shortcomings. First, the farthest ranging distance of ToF Lidar is limited by human eye safety standards. So far, there is no near-infrared ToF in the market that can detect objects with 5% reflectance at a distance of 200 m. Second, ToF Lidar is susceptible to environmental interference, including sunlight and the pulses emitted by other Lidar, which severely limits the popularization of ToF Lidar. Third, the ToF Lidar cannot directly measure the speed of objects. In the field of autonomous driving, the speeds of vehicles and pedestrians are critical, and being able to directly measure the speed information is crucial.

In this regard, LuminWave, Aeva, and other companies began to study high-performance FMCW Lidar. Such Lidar has a wide range of measuring, strong anti-interference ability, and can directly measure the speed according to the Doppler effect. The topic of “FMCW Lidar is the ultimate solution representing the future” is gradually being recognized by researchers. Aeva, LuminWave, and Aurora are already doing well in such products, and have received investment from many auto companies. Aeva expects to have the device in mass production by the end of 2023. Some parameters of FMCW Lidar in this section have been listed in Table 4.

In 2017, C.V. Poulton et al. used the optical phased array in Ref. [40] combined with the FMCW ranging method to produce a pure solid-state coherent Lidar on a silicon optical platform [63]. Different from the phased array in Ref. [40], the device divides the phase shifters into three different sets of electrical signals to achieve fine-tuning of the beam in order to account for any fabrication-induced phase noise. The system is controlled using nine copper electrical pads: three for the TX array, three for the RX array, two for the signal and bias of the balanced photoelectric detector, and one for the ground as shown in Figure 20. It has been demonstrated that the device can achieve velocity and distance measurement with 20 mm accuracy in the 2 m range.

In 2018, A. Martin et al. proposed a photonic integrated circuit-based FMCW Lidar system integrated on a 9 mm^2^ chip consisting of a cascade of Mach–Zehnder switching networks and a waveform calibration channel [64]. The passive waveguide is placed on the etched surface at a depth of 70 nm using shallow etching. This method reduces nonlinear absorption and allows higher power compared to deep etching, thus increasing the scanning range of the Lidar. As shown in Figure 21, the upper and lower arms of the Mach–Zehnder switch are folded in the heater section (left), and the profile of the heating section is shown on the right. In order to avoid additional transmission losses, the distance between the waveguide and the P-doped silicon heater is 2 μm. However, compared with the deeply etched waveguide at 600 nm, this etching method reduces the efficiency of the thermal phase shifter. After testing, the device can achieve moving object ranging of 60 m when the output power consumption is less than 5 mW.

In 2019, J. Riemensberger et al. demonstrated a soliton micro-comb massively parallel coherent Lidar based on ultra-low loss photonic chips [71]. The device transfers the chirp of a frequency-modulated Lidar source to multiple comb sidebands. Combined with the triangular frequency modulation of a narrow linewidth pump laser, it generates a massively parallel array of independent FMCW lasers. In this way, they generate 30 distinct channels, demonstrating both parallel distance and velocity measurements at an equivalent rate of three megapixels per second. They also accomplish 3D imaging of two sheets of paper with the EPFL university logo.

In 2019, X. Zhang et al. demonstrated an iterative learning pre-distortion laser sweep linearization method for FMCW Lidar [65]. The iterative learning controller can obtain the driving voltage waveform of the laser linear sweep frequency. Applying this method to the FMCW Lidar system, higher performance can be achieved without expensive linear lasers, complex linear setups, or heavy post-processing. An FMCW Lidar 3D imaging test based on this method is also presented. The distributed feedback laser is used because of its higher optical power. Figure 22a is the picture taken by the camera and Figure 22b,c is the point cloud image from the FMCW Lidar. Details such as toys, books, keyboards, and cups are restored. However, the desktop and mouse in the point cloud image are not very clear because of the specular reflection of the laser and the shallow angle of incidence.

In 2020, F. Zhang et al. proposed a dual-path FMCW Lidar that can simultaneously measure the speed and distance of an object [66]. The dual-path FMCW system uses a beam splitter to divide a laser into a measuring optical path and a reference optical path as shown in Figure 23. The former is used to illuminate the measured object and collect scattered light and the latter is used to provide the local oscillator light. The speed and distance of an object can be solved using the measurement data of two opposite scans in one period. The device does not require an additional optical path, reducing the complexity of the device and production cost. In addition, long fibers of known length are also used in the reference optical path to process the reference signal, eliminating the influence of frequency modulation nonlinearity on measurement. Based on the experimental results, the mean error for velocity measurement did not exceed 52 μm/s and the relative standard deviation was less than 2.5%.

In 2021, Z. Li et al. proposed a solid-state FMCW two-dimensional Lidar using virtual imaging phased array (VIPA). The system based on VIPA can achieve beam deflection only through dispersion [67]. As shown in Figure 24, the light source is a linear frequency-scanning laser. The collimator emits the laser beam and focuses on the VIPA through a cylindrical lens (CL). The transmission spectrum of a VIPA shows multiple resonance peaks spaced at an interval of a free spectral range ∆λ because of multiple beam interference, which leads to the light beams from the linear frequency-scanning laser with ∆λ wavelength difference having the same output angle along the y-direction. Then, the beam will be separated along the x-direction when it passes through the grating whose dispersion direction is orthogonal to VIPA’s. Thus, this system realizes 2D beam steering. The result shows that the coupling loss of VIPA is much less than most of the OPAs. However, due to the Gaussian shape diffraction envelope of VIPA, the system has certain transmitting power non-uniformity along the vertical field of view. Fortunately, the system adopts the FMCW ranging method based on MZI, which requires much lower power than ToF, which means that the non-uniformity can be compensated by increasing the source power.

In 2021, X. Cao et al. proposed an FMCW Lidar based on lens-assisted beam steering [68]. The device consists of an integrated SiO_2_ 1 × 16 switch chip, a 4 × 4 fiber optic array, and a lens, as shown in Figure 25a. Figure 25b shows the SiO_2_-based chip encapsulating the input and output of the single-mode fiber. The waveguide transmission loss is less than 0.05 dB/cm, ensuring adequate output and a long-ranging distance. Figure 25c shows the physical package diagram of the device. The Lidar operates at a 1550 nm, has 16 scan points, a steering step size of 0.35°, a steering angle of 1.05°, and a ranging distance of 80 m. The number of scanning points and scanning angles can be further increased by adjusting the parameters of the transmitting array and lens. This device demonstrates the potential of 2D lens-assisted beam steering Lidar in FMCW ranging.

In 2022, T. Baba et al. proposed a silicon optical FMCW Lidar chip based on a slow light-grating (SLG) scanner [69]. The research group has achieved a two-dimensional scanning angle of 40° × 8.8° and a beam divergence of 0.1. The FMCW Lidar based on SLG can realize point cloud image acquisition of 4928 points within 3~5 m, and the result of point cloud acquisition is shown in Figure 26. Figure 26a is the point cloud image of four rectangles with different distances, and Figure 26b is the point cloud image of a circular vertebra.

In 2022, K. Sayyah et al. produced two fully integrated silicon optical chips for the FMCW Lidar engine [70]. The chip integrates all electronic components, including the laser source, on a single silicon chip. Figure 27a shows the schematic diagram of the first-generation (Gen-1) chip. The chip uses the distributed Bragg reflector based on InP heterogeneous integration as the light source. The output waveguides of the MMI coupler feed dual-balanced Ge-on-Si photodiodes, resulting in the elimination of the laser relative intensity noise. The second generation (Gen-2) chip is shown in Figure 27b. Compared with the Gen-1 chip, the chip adopts the master oscillator power amplifier architecture as the laser source. It emits the laser directly into free space and the local oscillator source is coupled to the Si waveguide back from the facet of the laser in a special way. This design increases the power of the outgoing laser used to detect the target. The Si waveguide coupling efficiency of these two chips is up to 88% thanks to the heterogeneous integration technology of silicon photonic chips. Using the FMCW ranging method, the Gen-1 chip can achieve 28 m ranging when the transmitted light power is 2 mW, and the Gen-2 chip can achieve 75 m ranging with 80 mW. Both chips can be combined with off-chip beam steering elements to achieve compact 3D FMCW Lidar.

In 2023, H. Jang et al. demonstrated an FMCW Lidar that can measure distance and vibration information in real-time [30]. Thanks to the electro-optical external cavity diode laser used in this device, the system can generate uniform interference signals with a bandwidth of 1 GHz at a frequency sweep rate of 10 kHz without complicated post-processing or linearization devices. The device can obtain the vibration frequency and vibration rate of the target by tracking the temporal change in the beat frequency of the target. The system is used to map two speakers at different locations in three-dimensional space. The results are shown in Figure 28. Figure 28b shows the three-dimensional image of the distance measurement. In addition to the two speakers, the box and background are also highlighted. Figure 28c and Figure 28d, respectively, show the vibration frequency and vibration rate of the targets. Differently from Figure 28b, these two images highlight the parts under test that vibrate and mark the vibration information of different degrees with different colors. It has been demonstrated that the device can achieve spatial distance, vibration frequency, and vibration rate measurement and visualization processing. However, its anti-interference ability and stability need to be improved.

## 6. Conclusions

In this paper, we briefly describe the classification and ranging methods of Lidar and compare the characteristics of various Lidar. The research progress of silicon-based integrated Lidar, including optical phased array, optical switch array, and integrated FMCW Lidar, is introduced. At present, mechanical and MEMS Lidar are still the mainstream Lidar in the market that is gradually applied to the automatic drive. The silicon-based integrated Lidar is under research and development but has made much progress. More and more optical chips with wide FoV, high resolution, and low power consumption have been made.

Silicon-based phased array Lidar started earlier and has achieved high-performance 3D scanning. Silicon-based optical switch array Lidar has been developed for a short time, but remarkable achievements have been made. Compared with the former, it has the characteristics of a simple working mode and low power consumption, so it has greater potential. FMCW Lidar has stronger anti-interference ability and stability, can measure the velocity of objects, and has been recognized by most researchers. It will certainly occupy an important position in the future Lidar market. It is believed that with the progress of science and technology and the continuous improvement of semiconductor technology, silicon-based Lidar with full-solid states, low power consumption, and high precision performance are bound to be popular and bring convenience to people in their daily life.

## Figures and Tables

**Figure 1 sensors-23-05920-f001:**
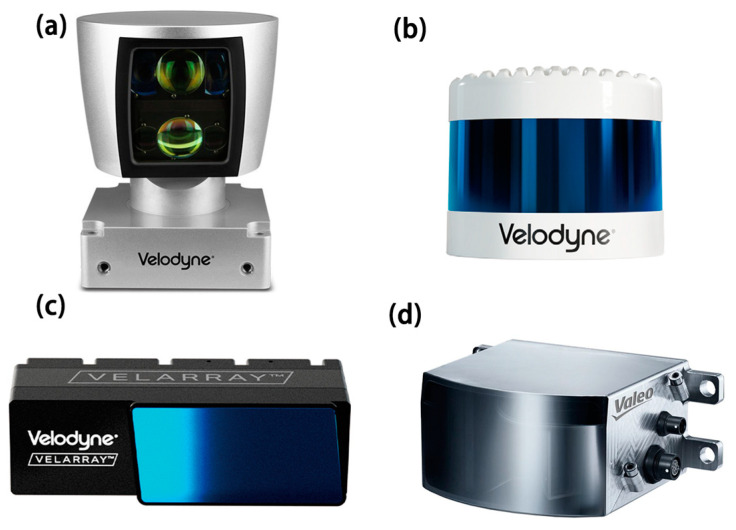
(**a**) HDL-64E from Velodyne. Size: 203.2 mm (outer diameter) × 281.94 mm (height). Weight: 13.2 kg; (**b**) VLS-128 from Velodyne. Size: 165.5 mm (outer diameter) × 141 mm (height). Weight: 3.5 kg; (**c**) Velarray H800 from Velodyne. Weight: <1 kg; (**d**) Scala GEN.1 from Valeo. Size: 133 mm × 101 mm × 70 mm. Weight: 625 g.

**Figure 2 sensors-23-05920-f002:**
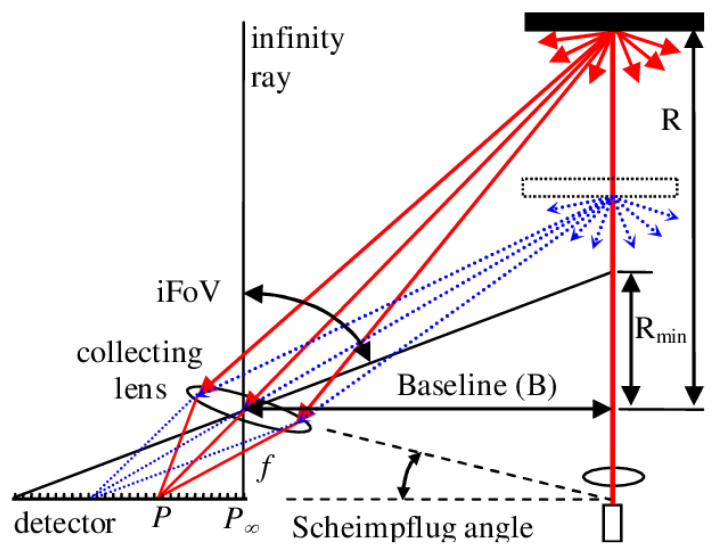
The schematic diagram of Triangular Ranging [21].

**Figure 4 sensors-23-05920-f004:**
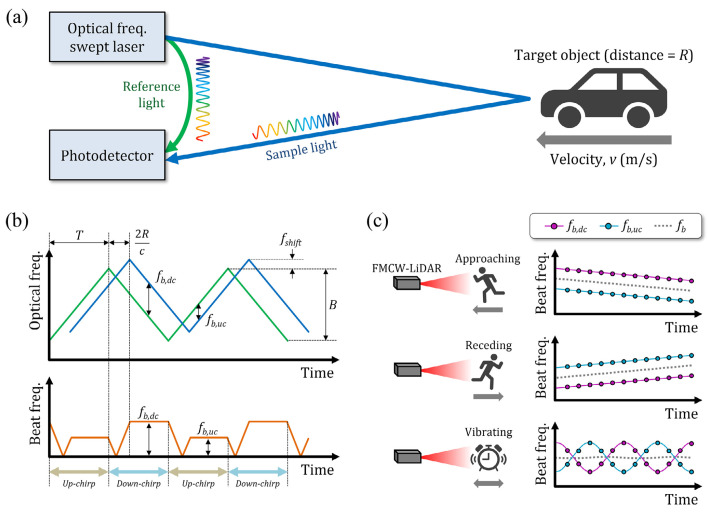
A conceptual diagram of FMCW Lidar. (**a**) The work diagram of FMCW Lidar. (**b**) Beat frequency of the reference signal (green) and sample signal (blue). (**c**) Beat frequency over time in the different moving cases, the blue and purple circles represent the measured peak beat frequencies [30].

**Figure 5 sensors-23-05920-f005:**
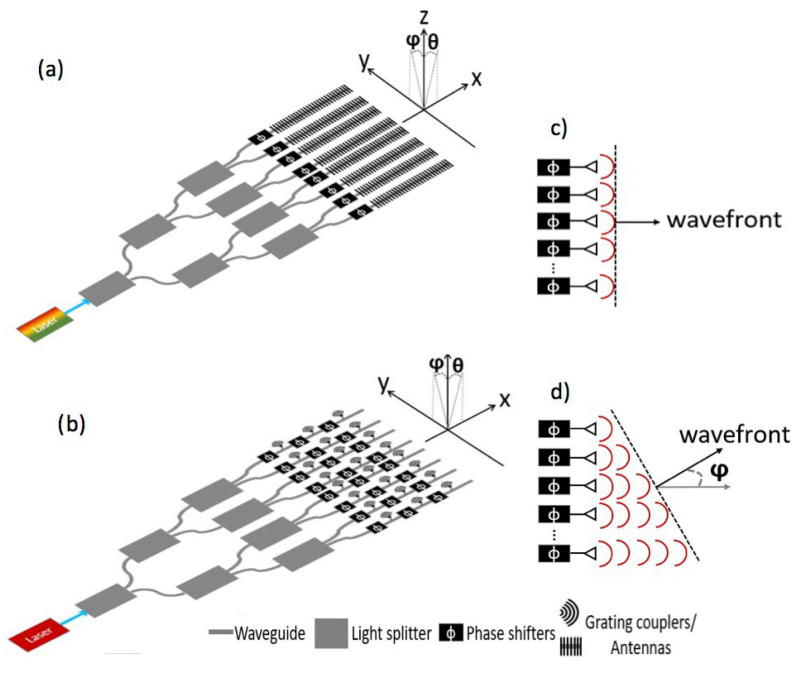
Schematic of optical phased array Lidar. (**a**) Optical steering in the longitudinal direction (θ) and lateral direction (φ) by 1D OPA. (**b**) Optical steering in directions by 2D OPA. (**c**) The emitted waves with no phase delay. (**d**) The emitted waves with phase delay [31].

**Figure 6 sensors-23-05920-f006:**
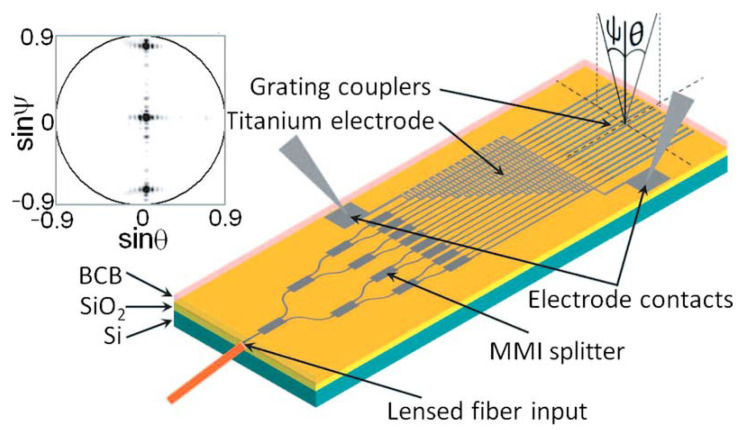
The optical phased array consists of 16 parallel grating couplers spaced 2 μm apart. θ: outcoupling angle along the waveguide axis. ψ: outcoupling angle perpendicular to the waveguide axis. The inset shows the far-field image [34].

**Figure 7 sensors-23-05920-f007:**
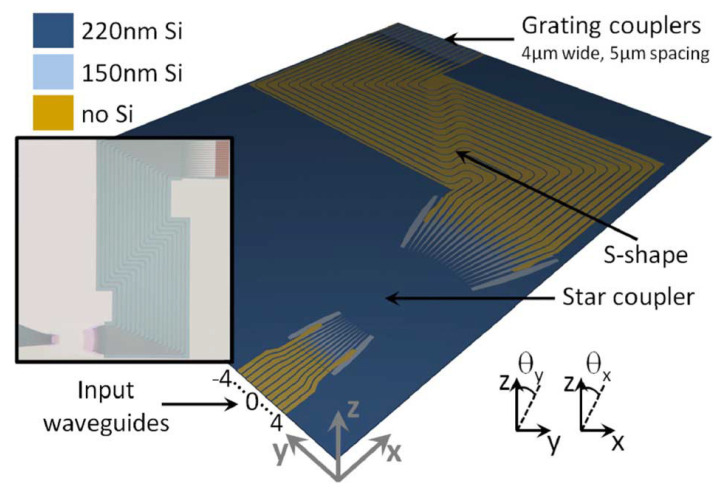
Virtual fabrication of a one-dimensional OPA on SOI. The inset shows a microscope image of the fabricated component before heater processing [38].

**Figure 8 sensors-23-05920-f008:**
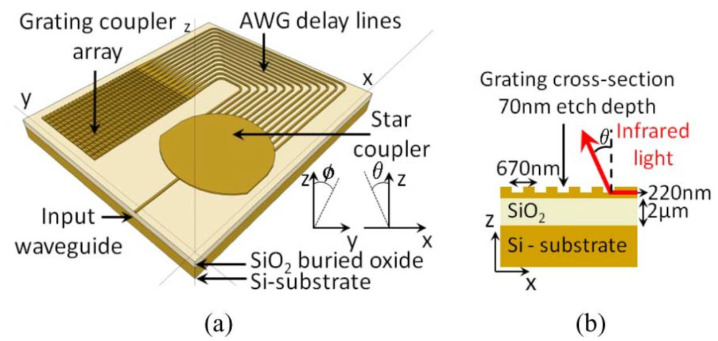
(**a**) Two-dimensional beam scanner on SOI. A high-order grating is used in the ϕ direction and a low-order grating is used in the θ direction. (**b**) Cross-section of the grating coupler array. The SOI wafer has a 2 μm buried oxide layer and a 220 nm silicon top layer [39].

**Figure 9 sensors-23-05920-f009:**
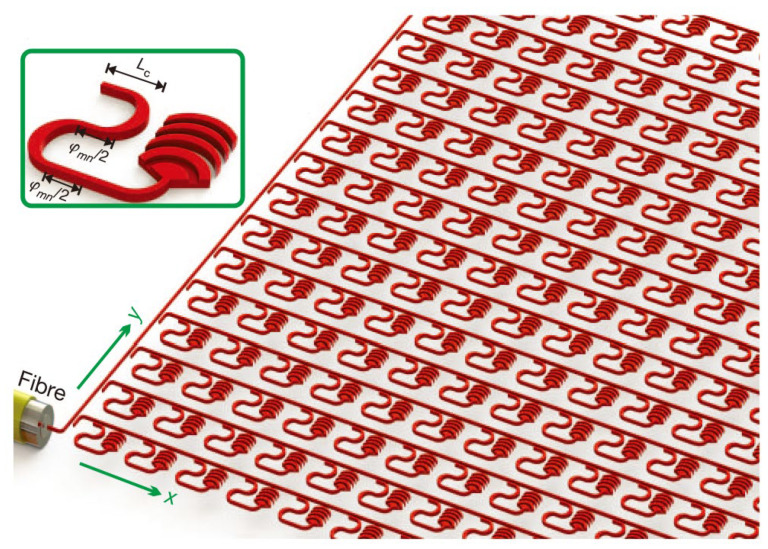
Schematic illustration of a 64 × 64 NPA system. The inset shows a diagram of a close-up view of one antenna unit cell. The length of directional coupler L_c_ can vary the coupling efficient and the two segments of the delay line can achieve a phase delay φ_mn_ [55].

**Figure 10 sensors-23-05920-f010:**
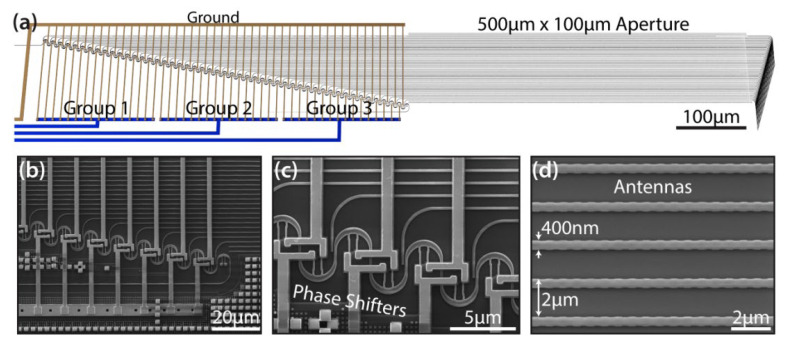
(**a**) SEM images of the (**b**) cascaded phase shifter architecture, (**c**) close-up of the thermal phase shifters, and (**d**) the full etch silicon grating-based antennas with a waveguide width of 400 nm and pitch of 2 µm [41].

**Figure 11 sensors-23-05920-f011:**
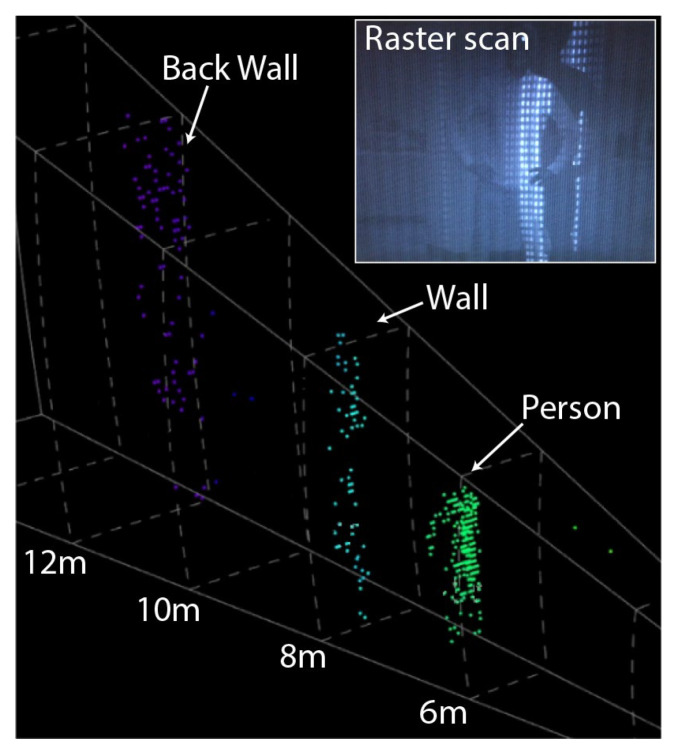
Real-time data from a 3D Lidar system consisting of raster scanning OPAs. Inset shows the scene being raster scanned [42].

**Figure 12 sensors-23-05920-f012:**
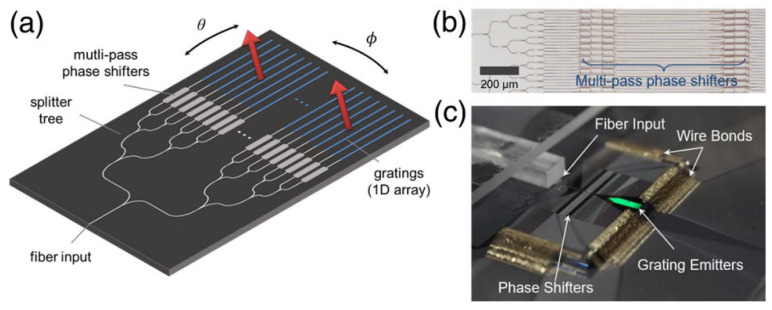
(**a**) Schematic (not to scale) of an optical phased array, showing out-of-plane beam emission (red arrows) and 2D steering. (**b**) Optical microscope image of the silicon waveguide layer of the fabricated chip. (**c**) Packaged device [44].

**Figure 13 sensors-23-05920-f013:**
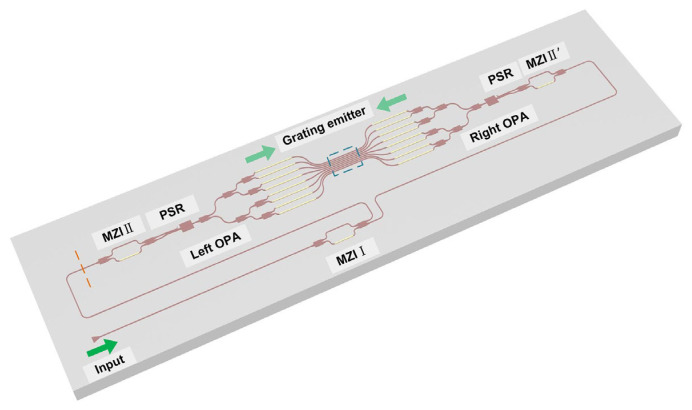
Schematic of the proposed bi-directional dual polarization multiplexed OPA. When the MZI I is on, the light injects into the grating emitter from the left side, otherwise from the right side [45].

**Figure 14 sensors-23-05920-f014:**
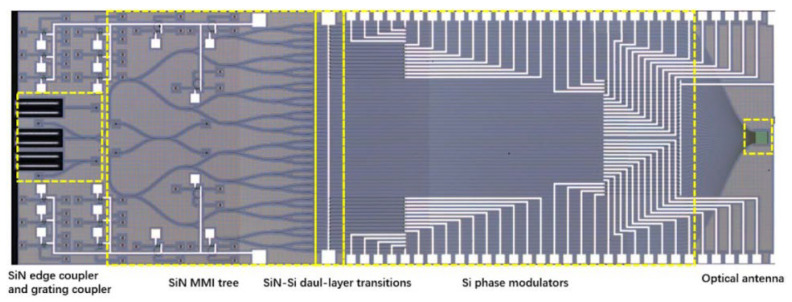
Optical micrograph of the proposed SiN-Si dual-layer optical phased array [48].

**Figure 15 sensors-23-05920-f015:**
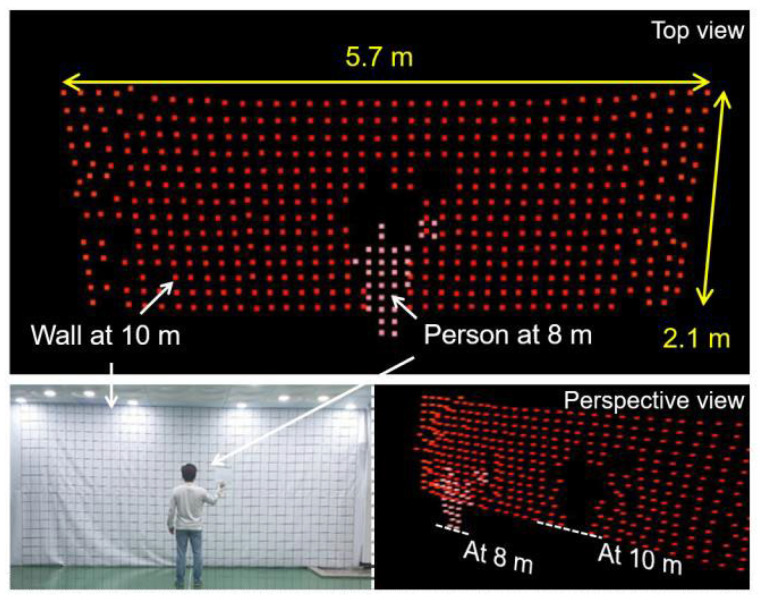
3D raster-scanned image detected by the Lidar system based on the ToF scheme [49].

**Figure 16 sensors-23-05920-f016:**
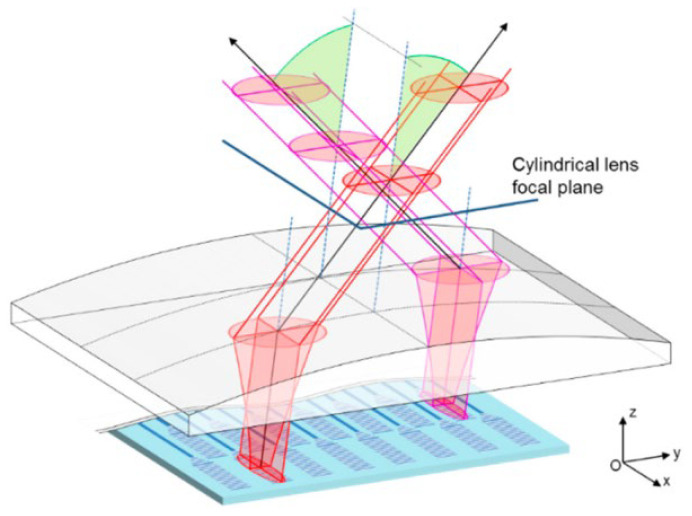
Schematic of the use of a cylindrical lens to compensate exit mode ellipticity and obtain sub-degree divergence in both the horizontal (x) and vertical (z) directions [57].

**Figure 17 sensors-23-05920-f017:**
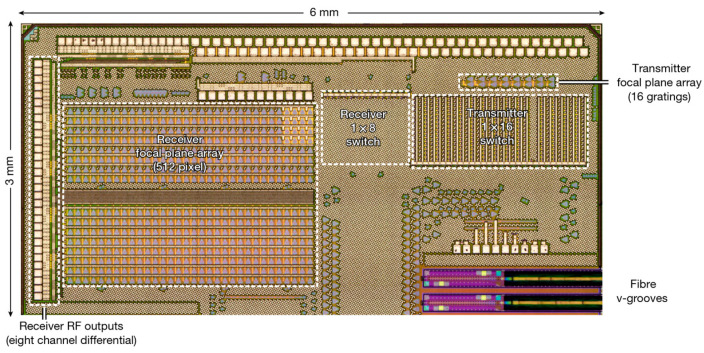
Optical micrograph of a demonstrator chip containing the receiver focal plane array and transmitter focal plane array [59].

**Figure 18 sensors-23-05920-f018:**
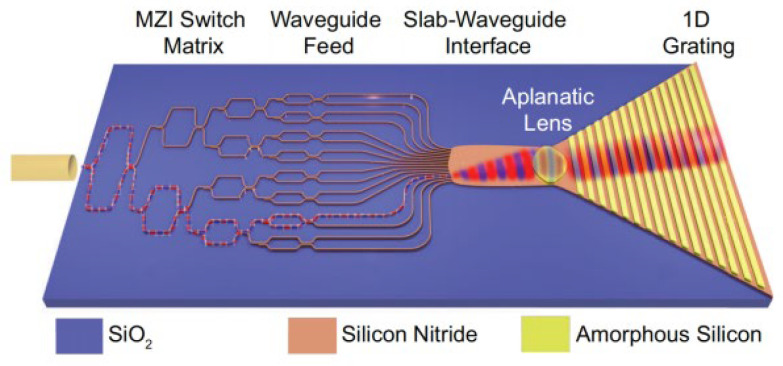
Schematic of the SiN switch array. An IR signal is coupled into the chip then through the MZI switch array, and finally is emitted from the lens [60].

**Figure 19 sensors-23-05920-f019:**
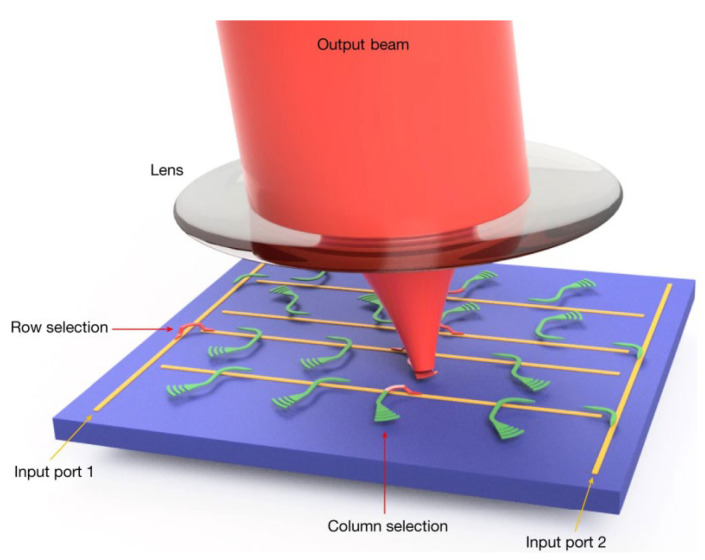
Perspective-view schematic of the 2D FPSA with the lens and output beam. The beam is coupled onto the chip and routed to the selected grating antenna by the column selection and row selection. The lens can ensure the collimation of the emitted light [62].

**Figure 20 sensors-23-05920-f020:**
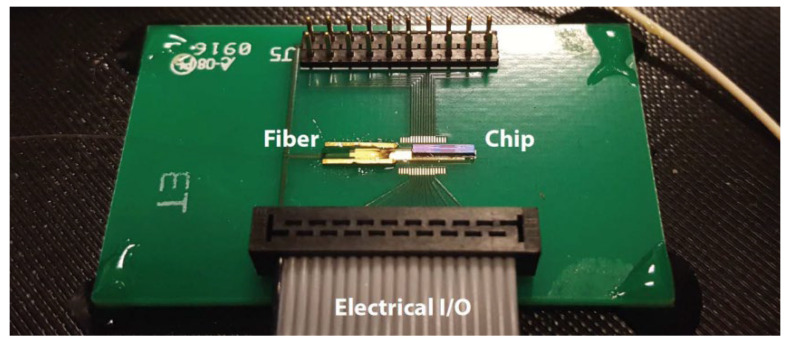
Packaged system with epoxied fiber [63].

**Figure 21 sensors-23-05920-f021:**
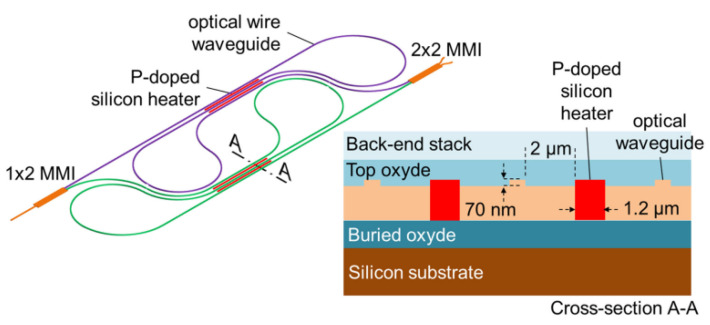
Schematic of a Mach–Zehnder switch cell and cross-section of the heater area. The upper and lower arms are folded around P-doped silicon heaters [64].

**Figure 22 sensors-23-05920-f022:**
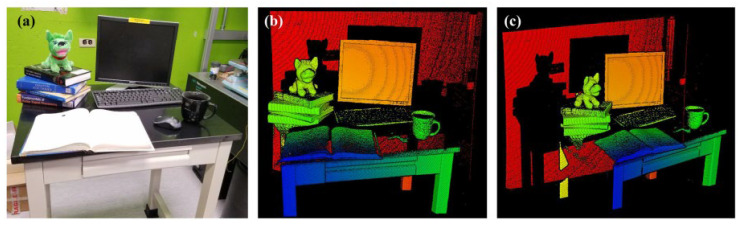
(**a**) Camera image of the scene. (**b**,**c**) Measured 3D point clouds [65].

**Figure 23 sensors-23-05920-f023:**
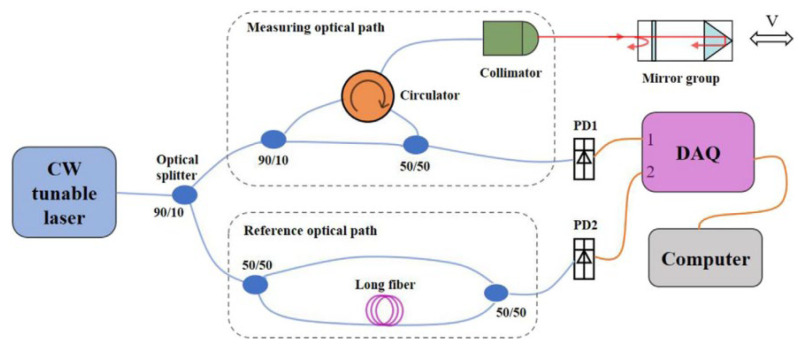
Schematic diagram of the dual-path FMCW Lidar system [66].

**Figure 24 sensors-23-05920-f024:**
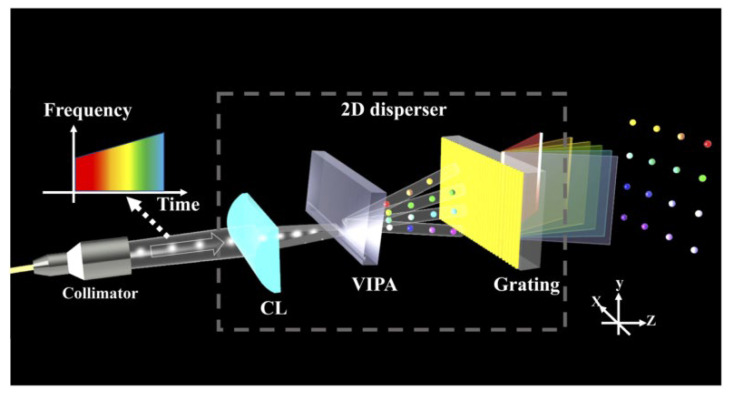
Schematic illustration of a 2D disperser. The different colors of light spots represent the different wavelengths [67].

**Figure 25 sensors-23-05920-f025:**
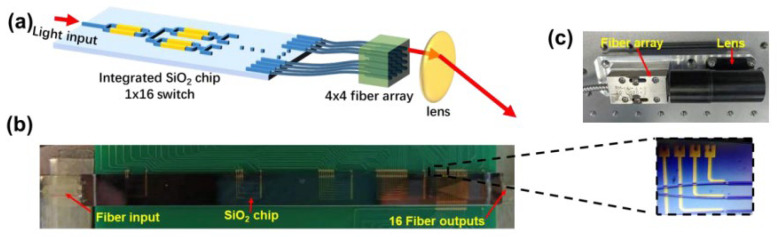
(**a**) Beam steering device. All the outputs of the chip are connected to the 4 × 4 fiber array to realize the conversion from 1D to 2D. (**b**) Photo of the SiO_2_ chip. Inset: zoom-in image. (**c**) Packaged fiber array and lens [68].

**Figure 26 sensors-23-05920-f026:**
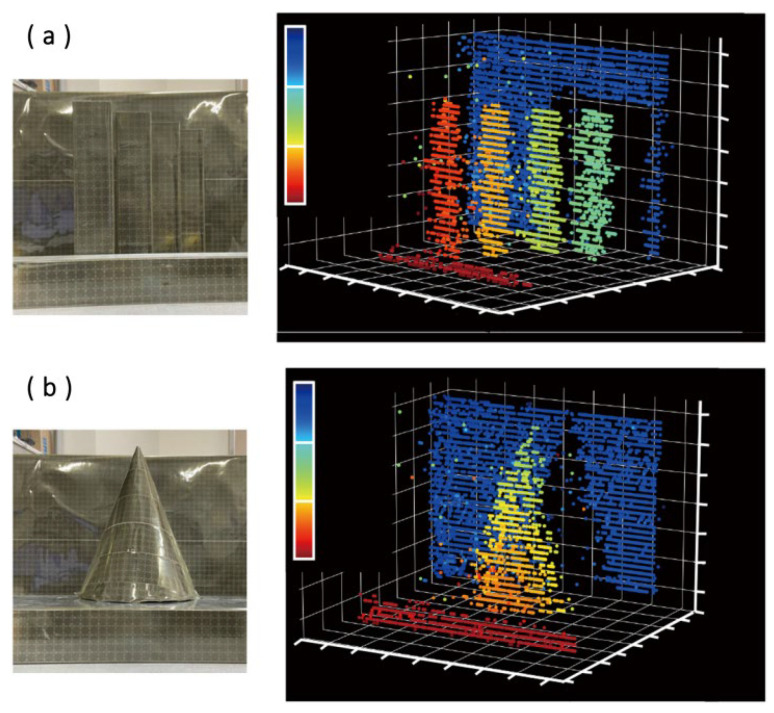
(**a**) Point cloud images of four rectangles with different distances. (**b**) Point cloud images of a circular vertebra [69].

**Figure 27 sensors-23-05920-f027:**
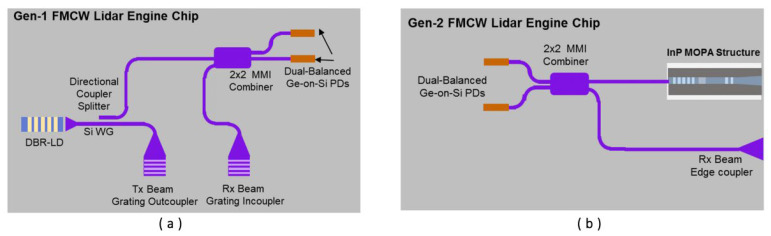
Schematic diagram of two generations of chips. (**a**) Gen-1. (**b**) Gen-2 [70].

**Figure 28 sensors-23-05920-f028:**
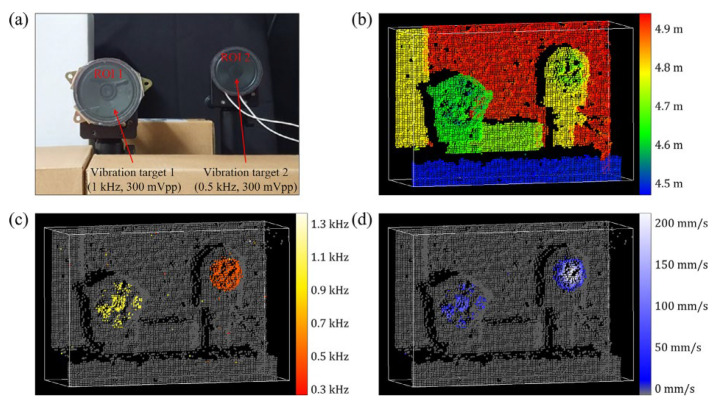
(**a**) Photograph of 3D visualization targets. (**b**) Measured image of the distance, (**c**) vibration frequency, and (**d**) maximum vibration velocity change [30].

**Table 1 sensors-23-05920-t001:** The differences between the above kinds of Lidar.

Variety	Advantage	Shortcoming	Example
Mechanical Lidar	Mature technology 360° horizontal FoV	Bulky [2] Poor stability High consumption	HDL-64E FoV: 360° × 26.8° resolution: 0.4° × 0.08–0.35° weight: 13.2 kg price: 28,064 $
MEMS Lidar	Light Lower cost [3,4] Simple in structure	Limited FoV Lack of stability	Velarray M1600 FoV: 120° × 32° resolution: 0.2° × 0.2–0.5° weight: <1 kg
Flash Lidar	Inexpensive Non-mechanical device	Susceptible to environment Limited FoV [10]	OUSTER-ES2 FoV: 13° × 26° resolution: 0.1° × 0.1° price: 100 $
Silicon-based OPA Lidar	Light Low cost High stability Compatible with CMOS process [12]	Still under development	-
Silicon-based Optical Switch Lidar	Simple construction Fast response speed [15] Compatible with silicon process	Still under development	-

**Table 2 sensors-23-05920-t002:** The parameters of the OPA Lidar in this section.

Reference	FoV (Degree)	Beam Width (Degree)	Platform Materials	New Finding (s)	Wavelength (nm)
[33]	0~3	0.25	SOI	irregular phased array	1550
[34]	2.3 × 14.1	-	SOI	thermo-optical and wavelength tuning	1550
[35]	-	-	SOI	high wavelength steering low antenna loss	1550
[36]	20 × 14	<1	SOI	Small phase error and background peak noise	1500 1550 1600
[37]	31.9	-	SOI	silicon nanomembrane with off-chip laser source	1550
[38]	23	1.27	SOI	waveguides of special structure to minimize the damage in sharp turns	1550
[39]	15 × 50	4.0	SOI	high steering efficiency	1500–1600
[40]	51	-	SOI	high speed beam steering	1550
[41]	46 × 36	-	SOI	utilize grouped cascaded phase shifters	1550
[15]	80 × 17	0.14	SOI	high resolution and wide beam steering angle	1260–1360
[42]	56 × 15	-	SOI	low power consumption and high directivity	1450–1640
[43]	18.5	0.15	SOI	the first Lidar system based on OPA	1550
[44]	70 × 6	-	SOI	cycle the light to achieve phase shifting	1525–1600
[45]	54.5 × 77.8	-	SOI	the bi-directional OPA with only one grating antenna array	1500–1600
[46]	-	0.089	SiN-Si	a solution to solve the trade-off between FoV and beam divergence	1550
[47]	51 × 28	0.02	hybrid III-V/Si waveguides	wide optical bandwidth high operating speed(1 GHz)	1550
[48]	48 × 14	-	SiN-Si	an ideal way to long range detection	1550
[49]	12 × 30	-	silicon nitride mixed polymer	high thermal and optical effect	-
[50]	96 × 14.4	1.9	SiN-Si	power processing capacity high thermal-optical modulation efficiency	1550

**Table 3 sensors-23-05920-t003:** The parameters of the optical switch array Lidar in this section.

Reference	FoV (Degree)	Beam Divergence (Degree)	Power Consumption	New Finding (s)	Wavelength (nm)
[56]	-	-	6.5 mW	using folded waveguide to reduce the switching power	1550
[57]	20 × 20	-	sub-μW level	phase shifter relying on piezoelectric transducer	-
[58]	-	-	logN *	high wavelength steering low antenna loss	1550
[59]	-	-	4 mW	large-scale coherent detector array with high accuracy	-
[60]	38.8 × 12	0.15	-	first optical planar-lens-enabled beam steering device	1550
[61]	40 × 4.4	0.15	-	designed a prism lens for beam steering and collimation	1550
[62]	70 × 70	0.050 × 0.049	-	128 × 128-element focal plan switch array with a wide FoV	1550

* N is the numbers of emitters.

**Table 4 sensors-23-05920-t004:** The parameters of FMCW Lidar in this section.

Reference	Detection Range	Accuracy	FoV (Degree)	New Finding (s)	Wavelength (nm)
[63]	2 m	20 mm	20	the first coherent Lidar with a silicon chip using OPA	1550
[64]	60 m	-	70	the scanning is accomplished by using collimation lenses	-
[65]	3 m	-	24 × 20	propose a new method for laser frequency sweep linearization	1550
[66]	205.595 mm	50 μm	-	a dual-path system with a simple structure and good nonlinear eliminate effect	1515–1565
[67]	1.8 m	0.5 mm	1.9 × 7.7	use the virtually imaged phased array to realize 2D beam steering	1500–1600
[68]	80 m	-	1.05	the scanning points and steering angle can be easily extended	1550
[69]	3–5 m	-	40 × 8.8	realize the point cloud image with 4928 pixels by slow-light gratings	1550
[70]	GEN-1 28 m GEN-2 75 m	GEN-1 28 cm GEN-2 16.7 cm	-	the first single chip-scale(include laser source) integrated FMCW Lidar	-
[30]	10 m	20.86 cm	-	realize the 3D-mapping of distance vibration frequency and vibration velocity	1467–1617

## Data Availability

Data are available on request due to restrictions, e.g., privacy or ethical.

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
