# Peer review of "Advances in Silicon-Based Integrated Lidar"

_sensors, 2023, doi:10.3390/s23135920_

Round 1
Reviewer 1 Report
The paper is a good review of an interesting field in constant expansion. Check the English. For example, some part of the following sentence seems to be inverted "Solid-state Lidar removes mechanical components from the device, including Flash Lidar, optical phased array, and optical switch array Lidar."
Is it possible to quantify the expected reduction of weigh, dimensions and costs made available by solid-state Lidar respect to mechanical Lidar?
Is it possible to add some columns, quantifying the achieved, by each technical solution, field of view, stability and resolution, to table 1?
Check the English. For example, some part of the following sentence seems to be inverted "Solid-state Lidar removes mechanical components from the device, including Flash Lidar, optical phased array, and optical switch array Lidar."
Author Response
Reviewer 1
- Comments and Suggestions for Authors:
The paper is a good review of an interesting field in constant expansion. Check the English. For example, some part of the following sentence seems to be inverted "Solid-state Lidar removes mechanical components from the device, including Flash Lidar, optical phased array, and optical switch array Lidar.".
Response from the authors:
Thank you for your helpful comment, we have check the English of the manuscript and rewritten the sentence above in lines 57-59.
- Comments and Suggestions for Authors:
Is it possible to quantify the expected reduction of weigh, dimensions and costs made available by solid-state Lidar respect to mechanical Lidar?
Response from the authors:
We have listed the weight and dimensions of the mechanical Lidar in lines 53-56. And the dimensions of the solid-state Lidar have been mentioned in sections 3-5 for example in line 611, we introduce the FMCW Lidar system is integrated on a 9 mm2 chip and in line 324, the OPA Lidar chip is fabricated in a 300 mm facility.
- Comments and Suggestions for Authors:
Is it possible to add some columns, quantifying the achieved, by each technical solution, field of view, stability and resolution, to table 1?
Response from the authors:
We have add the concrete examples of mechanical, MEMS and flash Lidar in Table 1 and listed the FoV, resolution, weight and price of these commercial Lidar. As for the OPA and optical switch array Lidar, there is no commercial product in the market, but we have listed the performance parameter of some chips under development in tables 2-4.
Reviewer 2 Report
Great review paper which summarizes the several Lidar technologies. I enjoyed to read the paper, as it goes into the descriptions of the Lidar technologies following the timelaps of more recent works. I have just a few of minor comments:
- Page 1. At sentence “the angular resolution of Lidar is no less than 0.1mard”. What is “0.1mard”? Please check it
- Along the entire manuscript there are a number of typos, e.g. many missing spaces. Please check for typos along the manuscript. A automatic corrector should identify them immediately.
- Even if the paper reports several citations, the section 1 and section 2 misses proper citations. There are several statements that are not supported by references.
- Captions of figures are too short in general. It could be usefull to insert in the caption more descriptive details about what is reported in the figures.
- Please re-check the sizes of figures, as some plots reported in the figures are not very readable due to the small dimensions, e.g., Fig 26(c)
Engligh is relatively good, but several typos are Along the entire manuscript, e.g. many missing spaces. Please check for typos along the manuscript. A automatic corrector should identify them immediately.
Author Response
Reviewer 2
- Comments and Suggestions for Authors:
Page 1. At sentence “the angular resolution of Lidar is no less than 0.1 mard”. What is “0.1 mard”? Please check it.
Response from the authors:
Thank you for your helpful comment, we have corrected the word “mard” to “mrad” in line 29.
- Comments and Suggestions for Authors:
Along the entire manuscript there are a number of typos, e.g. many missing spaces. Please check for typos along the manuscript. An automatic corrector should identify them immediately.
Response from the authors:
We have corrected the typos and add the spaces when it needed in this manuscript.
- Comments and Suggestions for Authors:
Even if the paper reports several citations, the section 1 and section 2 misses proper citations. There are several statements that are not supported by references.
Response from the authors:
We have checked and corrected the references and citations in the section 1 and section 2.
- Comments and Suggestions for Authors:
Captions of figures are too short in general. It could be useful to insert in the caption more descriptive details about what is reported in the figures.
Response from the authors:
We have add details in some figures in this manuscript such as figure 1, figure 4, figure 5, figure 6, figure 7, figure 8, figure 9, figure 13, figure 18, figure 19, figure 21 and figure 25.
- Comments and Suggestions for Authors:
Please re-check the sizes of figures, as some plots reported in the figures are not very readable due to the small dimensions, e.g., Fig 26 (c).
Response from the authors:
We have adjusted the size of some figures in this manuscript, such as figure 17 and figure 25 (figure 26 in the previous edition).
Reviewer 3 Report
This is the review report for a review manuscript titled "Advances on Silicon-based Integrated Lidar" by Mingxuan hu et al.
The manuscript has provided background and technical working principles for the Lidar technology. Three major focuses of the Lidar technology are highlighted - silicon-based optical phased array Lidar, silicon-based optical switch array Lidar, and Integrated FMCW Lidar.
Some comments for the authors to consider:
In Section 1 Introduction:
1) In the second paragraph, is there any reason why Velodyne's Lidar product is mentioned? Are there other leading manufacturers by market cap or revenue that the reader should know? It was also noted that there is no other commercial example for the other variety of Lidar in this section.
2) Figures 1 and 2. I would suggest to include the size dimension (L x W x H as overlay) of the commercial Lidar item to illustrate their bulk size further. Perhaps the two figures' images could be condensed into a single figure as the physical appearance of the product does not matter to its general usage.
3) Table 1. It will be better to put the references again in the table for readers to quickly relate to past reports. As a new column or inside the existing variety column.
In section 2 Lidar Ranging Methods:
4) The theory portion of the manuscript seems to be quite short with one equation on Page 5 for a review manuscript. It is suggested to provide more theoretical finding
In sections 3 (Silicon-based Optical Phased Array Lidar), 4 (Silicon-based Optical Switch Array Lidar), and 5 (Integrated FMCW Lidar):
5) The authors could consider grouping the silicon Lidar into subsections (i.e. 3.1, 3.2,... 4.1, 4.2,..., 5.1, 5.2,... ) instead of going by the years sequentially and discussed each paper individually. In other words, the review should organise and summaries the past reports. As an example, 3. Silicon-based Optical Phased Array Lidar, can be subsection into 3.1 Grating based phase shifters, 3.2 Metamaterial / Nanophotonics based, 3.3 2D material based, etc. depending on how the authors can propose to group it.
6) An additional summary table to list all the different advances (e.g. group[reference], structure/working principle, new finding(s), operating wavelength, steering step size, angle, range, etc.) at the start of each of the major sections (3, 4, and 5) will provide the reader with clearer picture of the development landscape.
Other non-technical comments:
7) There are some inconsistencies in the manuscript. Examples, page 7 paragraph 2 line 4 (spacing between SI unit, and degree word instead of degree symbol °), and page 8 line 4 (extra spaces).
The English language is fluent but can be improved. The manuscript's literature should be summarised based.
Author Response
Reviewer 3
- Comments and Suggestions for Authors:
In the second paragraph, is there any reason why Velodyne's Lidar product is mentioned? Are there other leading manufacturers by market cap or revenue that the reader should know? It was also noted that there is no other commercial example for the other variety of Lidar in this section.
Response from the authors:
Thank you for your helpful comment. The reason why Velodyne’s Lidar product is mentioned is that this company has a high market share in the Lidar area. Similarly, the Valeo’s product is mentioned too in line 47.
- Comments and Suggestions for Authors:
Figures 1 and 2. I would suggest to include the size dimension (L x W x H as overlay) of the commercial Lidar item to illustrate their bulk size further. Perhaps the two figures' images could be condensed into a single figure as the physical appearance of the product does not matter to its general usage.
Response from the authors:
We have added the size and weight information of mechanical Lidar and MEMS Lidar in line 53, and condensed the figure 1 and figure 2 into a single figure.
- Comments and Suggestions for Authors:
Table 1. It will be better to put the references again in the table for readers to quickly relate to past reports. As a new column or inside the existing variety column.
Response from the authors:
We have put the reference in the table 1.
- Comments and Suggestions for Authors:
The theory portion of the manuscript seems to be quite short with one equation on Page 5 for a review manuscript. It is suggested to provide more theoretical finding.
Response from the authors:
We have added the theory part in section two and added more equations in this section.
- Comments and Suggestions for Authors:
The authors could consider grouping the silicon Lidar into subsections (i.e. 3.1, 3.2,... 4.1, 4.2,..., 5.1, 5.2,... ) instead of going by the years sequentially and discussed each paper individually. In other words, the review should organize and summaries the past reports. As an example, 3. Silicon-based Optical Phased Array Lidar, can be subsection into 3.1 Grating based phase shifters, 3.2 Metamaterial / Nanophotonics based, 3.3 2D material based, etc. depending on how the authors can propose to group it.
Response from the authors:
We have subsection the section 3 into 3.1 OPA Lidar based on silicon on insulator (line 203) and 3.2 OPA Lidar based on hybrid material (line 386), and the section 4 into 4.1 low-consumption optical switch array Lidar (line 480) and 4.2 wide-FoV optical switch array Lidar (line 528).
- Comments and Suggestions for Authors:
An additional summary table to list all the different advances (e.g. group[reference], structure/working principle, new finding(s), operating wavelength, steering step size, angle, range, etc.) at the start of each of the major sections (3, 4, and 5) will provide the reader with clearer picture of the development landscape.
Response from the authors:
We have listed table 2, table 3, table 4 to summarize the major content in sections 3, 4 and 5.
- Comments and Suggestions for Authors:
There are some inconsistencies in the manuscript. Examples, page 7 paragraph 2 line 4 (spacing between SI unit, and degree word instead of degree symbol °), and page 8 line 4 (extra spaces).
Response from the authors:
We have check the spaces in the manuscript, than deleted the extra spaces and add the missing spaces. And we instead the degree of the symbol °.
Round 2
Reviewer 2 Report
Thank you for following my previous comments. The paper has gained quality and reads smoothly. Thank you for this interesting review.
I have a very minor comment:
MI. Please check the text in the caption of Fig. 13, as there is a typo: "When the MZI I on, the light...".
Author Response
Thank you for your comment. We have revised this caption.
Reviewer 3 Report
This is the second review report for a review manuscript titled "Advances on Silicon-based Integrated Lidar" by Mingxuan Hu et al.
The authors have provided their responses in the Author's notes to address all the comments. I would like to thank the authors for taking in my comments positively from the first review report, for their consideration, and kindly amended their manuscript. The authors have also summarised their findings into table form.
After reviewing the manuscript again, I have no other comment on the manuscript.
Author Response
Thank you for your comment.